# Clinical significance of non-thyroidal illness syndrome on disease activity and dyslipidemia in patients with SLE

Xin Zhang[1], Lirong Liu[1,2], Xiaolei Ma[1], Wei Hu[3], Xue Xu[1], Saisai Huang[1], Bingzhu Hua[1], Hong Wang[1], Zhiyong Chen●[1]*, Lingyun Sun[1]*

1 Department of Rheumatology and Immunology, The Affiliated Drum Tower Hospital, Nanjing University Medical School, Nanjing, China, 2 Department of Rheumatology and Immunology, The first Hospital of Changshu, Changshu, China, 3 Department of Clinical Laboratory, The Affiliated Drum Tower Hospital, Nanjing University Medical School, Nanjing, China

* lingyunsun2012@163.com (LS); chainzhiyong@163.com (ZC)

## Abstract

### Objectives

Nonthyroidal illness syndrome (NTIS), also known as low triiodothyronine (T3) syndrome, frequently affects patients with systemic lupus erythematosus (SLE) and may affect lipid metabolism. Dyslipidemia is highly prevalent and associated with the long-term prognosis of SLE. The aim of the present study was to explore the clinical significance of NTIS on disease activity and dyslipidemia in patients with SLE.

### Methods

Clinical and laboratory data were collected retrospectively from 223 patients with SLE. The correlation between free triiodothyronine (FT3), SLE disease activity, and lipid profiles were estimated. The correlation coefficient (r) was calculated using a Pearson's regression model. Univariate and multivariate logistic regression analyses were performed to identify the risk factors for dyslipidemia in SLE.

### Results

Serum FT3 levels were negatively correlated with the levels of 24 h urine protein (UP), blood urea nitrogen (BUN), creatinine (Cr) and SLE disease activity index (SLEDAI) (all $p < 0.001$) in NTIS patients but not in euthyroid patients. ApoB/ApoA1 was significantly correlated with SLEDAI ($p < 0.01$) in NTIS patients and CRP ($p < 0.001$) and ESR ($p < 0.01$) in euthyroid patients. A multivariate analysis revealed that only FT3 exhibited an independent negative association with dyslipidemia ($P = 0.01$; OR = 0.48; 95% CI 0.27–0.85).

### Conclusion

NTIS frequently occurs in patients with SLE. Low FT3 is associated with disease activity in SLE patients complicated with NTIS. Low FT3 is an independent risk factor for dyslipidemia in patients with SLE.

**Data Availability Statement:** All relevant data are within the paper.

**Funding:** This study was supported by Nanjing Medical Science and Technique Development

Foundation (Z.C., ZKX16039). The funders had no role in study design, data collection and analysis, decision to publish, or preparation of the manuscript.

**Competing interests:** The authors have declared that no competing interests exist.

## Introduction

Systemic lupus erythematosus (SLE) is an autoimmune disease characterized by autoantibody production in women of childbearing age [1]. Dyslipidemia, characterized as increased total cholesterol (TC), triglyceride (TG), low-density lipoprotein (LDL) and/or decreased high-density lipoprotein (HDL) levels in the serum, is prevalent in SLE patients with an incidence ranging between 18.1% and 75% [2]. Importantly, dyslipidemia is associated with disease activity (e.g., kidney damage and cardiovascular disease), and is closely related to the long-term prognosis of SLE patients [2–4]. Nonthyroidal illness syndrome (NTIS; also known as low triiodothyronine [T3] syndrome or euthyroid sick syndrome) is also prevalent in SLE and characterized by decreased serum T3, normal to low thyroxine (T4), and thyroid-stimulating hormone (TSH) levels [5–7]. The aim of the present study was to explore the prevalence and clinical significance of NTIS in a Chinese cohort with SLE. We hypothesized that as a critical metabolic hormone, low FT3 levels may be associated with disease activity and the lipid profile in patients with SLE.

## Materials and methods

### Participants and data collection

A single-center, observational and cross-sectional study was performed. A total of 271 patients hospitalized at the Affiliated Drum Tower Hospital of Nanjing University Medical School (Nanjing, China) from Dec. 2015 to Apr. 2017 were recruited consecutively (if informed consent was obtained). Clinical and laboratory data were collected retrospectively from medical records. All patients included in the analysis were informed and agreed to participate in this research. The study protocol was approved by the Ethics Review Committee of the Affiliated Drum Tower Hospital.

All SLE patients fulfilled the American College of Rheumatology (ACR) classification criteria for SLE (2009). Forty-eight patients who had a history of familial hyperlipidemia and/or thyroid disease, diabetes mellitus, and/or other rheumatic diseases, and those who took lipid-lowering agents or thyroid medications were excluded. Eventually, 223 patients were enrolled in this study. The SLE disease activity index (SLEDAI) was assessed by qualified rheumatology specialists (B.H., H.W. and Z.C.). Dyslipidemia was defined according to the Chinese guidelines for the management of dyslipidemia in adults (i.e., $TC \geq 5.2$ mmol/L; $TG \geq 2.3$ mmol/L; $HDL < 1.0$ mmol/L; $LDL \geq 4.1$ mmol/L) [8].

The thyroid hormones, including free T3 (FT3), free thyroxine (FT4), and TSH were measured using a commercially available electrochemiluminescence (ECLIA) kit (Roche). The normal ranges were considered to be: 3.1–6.8 pmol/L for FT3; 12–22 pmol/L for FT4; and 0.27–4.2 mU/L for TSH. The following criteria for NTIS were applied in this study: FT3 less than 3.1 pmol/L; normal to low FT4; and normal TSH levels [9].

### Statistical analysis

All continuous variables are presented as the mean ± standard deviation (SD). Categorical variables were compared using a chi-square test. Independent-sample *t*-tests were used to compare continuous normally distributed variables. Values that were not normally distributed were evaluated using a Mann-Whitney U test. The correlation coefficient (r) was calculated using a Pearson's regression model. Variables for the risk of dyslipidemia were identified by univariate and multivariate logistic regression analyses using the following parameters: sex, age at entry into the study, the levels of FT3, FT4, TSH, creatinine (Cr), blood urea nitrogen (BUN), 24 h urine protein (24 h UP), C-reactive protein (CRP), erythrocyte sedimentation

rate (ESR), SLEDAI, and dose of glucocorticoids (calculated using the equivalent dose of methylprednisolone) at the time of sampling. The results were presented as odds ratios (ORs) with 95% confidence intervals (95% CIs). A $p$ value < 0.05 was considered statistically significant. All statistical analyses were performed using SPSS software (ver. 24.0, SPSS Inc, Chicago, IL) or Prism (GraphPad Software, San Diego, CA).

## Results

As shown in Table 1, the incidence of NTIS in SLE patients in this study was 58.7% (131/223). There is no statistically significant difference between SLE patients in the euthyroid and NTIS groups regarding age, sex, and positive rates of anti-nuclear antibodies (ANA), anti-Sm antibodies, anti-dsDNA antibodies, and the duration of glucocorticoid treatment. The levels of FT3 (2.3 ± 0.6 pmol/L vs. 4.1 ± 0.7 pmol/L), FT4 (13.6 ± 3.3 pmol/L vs. 15.9 ± 2.3 pmol/L), and TSH (1.4 ± 0.9 mU/L vs. 1.8 ± 1.0 mU/L) were significantly decreased in patients in the NTIS group compared to those in the euthyroid group (all $p$ < 0.001). Clinical indexes, including 24 h UP (2340.9 ± 3217.4 mg vs. 752.1 ± 863.6 mg; $p$ < 0.001), BUN (7.8 ± 5.4 mmol/L vs. 5.0 ± 2.0 mmol/L; $p$ < 0.001), Cr (76.1 ± 69.7 μmol/L vs. 55.0 ± 34.7 μmol/L; $p$ < 0.01), uric acid (331.6 ± 143.5 μmol/L vs. 295.1 ± 109.0 μmol/L; $p$ < 0.05), CRP (21.5 ± 38.9 mg/L vs.

**Table 1. Comparison of the characteristics of SLE patients with and without NTIS.**

| Characteristic | SLE | | *P* value |
| --- | --- | --- | --- |
| | **Without NTIS (n = 92)** | **With NTIS (n = 131)** | |
| Age, years | 36.9 ± 12.6 | 36.7 ± 13.8 | NS |
| Male/Female, no. | 10/82 | 6/125 | NS |
| ANA positive, no. (%) | 91 (99) | 116 (89) | NS |
| Anti-Sm positive, no. (%) | 21 (22.8) | 33 (25.2) | NS |
| Anti-dsDNA positive, no. (%) | 30 (32.6) | 50 (38.2) | NS |
| Duration of glucocorticoid treatment, months | 12.3 ± 2.4 | 14.5 ± 2.9 | NS |
| FT3, pmol/L | 4.1 ± 0.7 | 2.3 ± 0.6 | < 0.001 |
| FT4, pmol/L | 15.9 ± 2.3 | 13.6 ± 3.3 | < 0.001 |
| TSH, mU/L | 1.8 ± 1.0 | 1.4 ± 0.9 | < 0.001 |
| 24 h UP, mg | 752.1 ± 863.6 | 2340.9 ± 3217.4 | < 0.001 |
| BUN, mmol/L | 5.0 ± 2.0 | 7.8 ± 5.4 | < 0.001 |
| Cr, μmol/L | 55.0 ± 34.7 | 76.1 ± 69.7 | < 0.01 |
| UA, μmol/L | 295.1 ± 109.0 | 331.6 ± 143.5 | < 0.05 |
| CRP, mg/L | 11.9 ± 23.2 | 21.5 ± 38.9 | < 0.05 |
| ESR, mm/h | 32.8 ± 27.6 | 55.1 ± 33.1 | < 0.001 |
| SLEDAI | 4.0 ± 3.6 | 8.5 ± 4.6 | < 0.01 |
| TG, mmol/L | 1.46 ± 0.8 | 1.9 ± 1.3 | < 0.01 |
| TC, mmol/L | 4.3 ± 0.9 | 4.9 ± 1.9 | < 0.01 |
| HDL, mmol/L | 1.2 ± 0.6 | 1.0 ± 0.4 | < 0.01 |
| LDL, mmol/L | 2.3 ± 0.7 | 2.8 ± 1.4 | < 0.01 |
| ApoA1, g/L | 1.0 ± 0.3 | 1.0 ± 0.5 | NS |
| ApoB, g/L | 0.8 ± 0.2 | 1.1 ± 0.5 | < 0.001 |
| ApoB/ApoA1 | 0.7 ± 0.02 | 1.1 ± 0.03 | < 0.001 |

Values are presented as the mean ± SD unless otherwise indicated; ANA = anti-nuclear antibody; FT3 = free triiodothyronine; FT4 = free thyroxine; TSH = thyroid stimulating hormone; 24 h UP = 24 h urine protein; BUN = blood urea nitrogen; Cr = creatinine; UA = uric acid; CRP = C reactive protein; ESR = erythrocyte sedimentation rate; SLEDAI = systemic lupus erythematosus disease activity index; TG = triglyceride, TC = total cholesterol, HDL = high-density lipoprotein, LDL = low-density lipoprotein, ApoA1 = apolipoprotein A1, ApoB = apolipoprotein B; NS = not significant.

11.9 ± 23.2 mg/L; *p* < 0.05), ESR (55.1 ± 33.1 mm/h vs. 32.8 ± 27.6 mm/h; *p* < 0.001), and SLE-DAI (8.5 ± 4.6 vs. 4.0 ± 3.6; *p* < 0.01) displayed an obvious increase in SLE patients with NTIS compared to the euthyroid group. Regarding lipid profiles, we found that SLE patients with NTIS had a substantially higher TG (1.9 ± 1.3 mmol/L vs. 1.46 ± 0.8 mmol/L; *p* < 0.01), TC (4.9 ± 1.9 mmol/L vs. 4.3 ± 0.9 mmol/L; *p* < 0.01), LDL (2.8 ± 1.4 mmol/L vs. 2.3 ± 0.7 mmol/L; *p* < 0.01), apolipoprotein B (ApoB) (1.1 ± 0.5 g/L vs. 0.8 ± 0.2 g/L; *p* < 0.001), ApoB/apolipoprotein A1 (ApoA1) (1.1 ± 0.03 vs. 0.7 ± 0.02; *p* < 0.001), as well as considerably lower HDL (1.0 ± 0.4 mmol/L vs. 1.2 ± 0.6 mmol/L; *p* < 0.01) levels compared with patients in the euthyroid group. However, the ApoA1 levels were comparable between these two groups.

We further explored the relationship between the associated parameters and lipid profiles for thyroid hormones and SLE disease activity in patients with or without NTIS. A regression analysis revealed that the FT3 levels were negatively correlated with 24 h UP (*r* = -0.301; *p* < 0.001; Fig 1A), BUN (*r* = -0.325; *p* < 0.001; Fig 1B), Cr (*r* = -0.298; *p* < 0.001; Fig 1C), as well as SLEDAI (*r* = -0.313; *p* < 0.001; Fig 1F). However, these correlations were not observed in patients without NTIS (Fig 2A–2C and 2F). In addition, FT3 were negatively correlated with CRP in both NTIS patients (*r* = -0.200; *p* < 0.05; Fig 1D) and euthyroid patients (*r* = -0.254; *p* < 0.05; Fig 2D), and were negatively correlated with ESR (*r* = -0.27; *p* < 0.05; Fig 2E) in NTIS patients rather than in euthyroid patients (Fig 1E). These results indicate that low levels of FT3 are closely related to a higher SLE disease activity primarily in NTIS patients rather than in euthyroid patients.

There is increasing evidence to show that the atherogenic ApoB/ApoA1 ratio is an independent prospective predictor of cardiovascular events both in the general population and in patients with rheumatic diseases [10–13]. We also found that the ApoB/ApoA1 ratio was significantly associated with SLEDAI (*r* = 0.251; *p* < 0.01; Fig 3E) but not with CRP, ESR, Cr, or 24 h UP (Fig 3A–3D) in NTIS patients. In euthyroid patients, the ApoB/ApoA1 ratio was

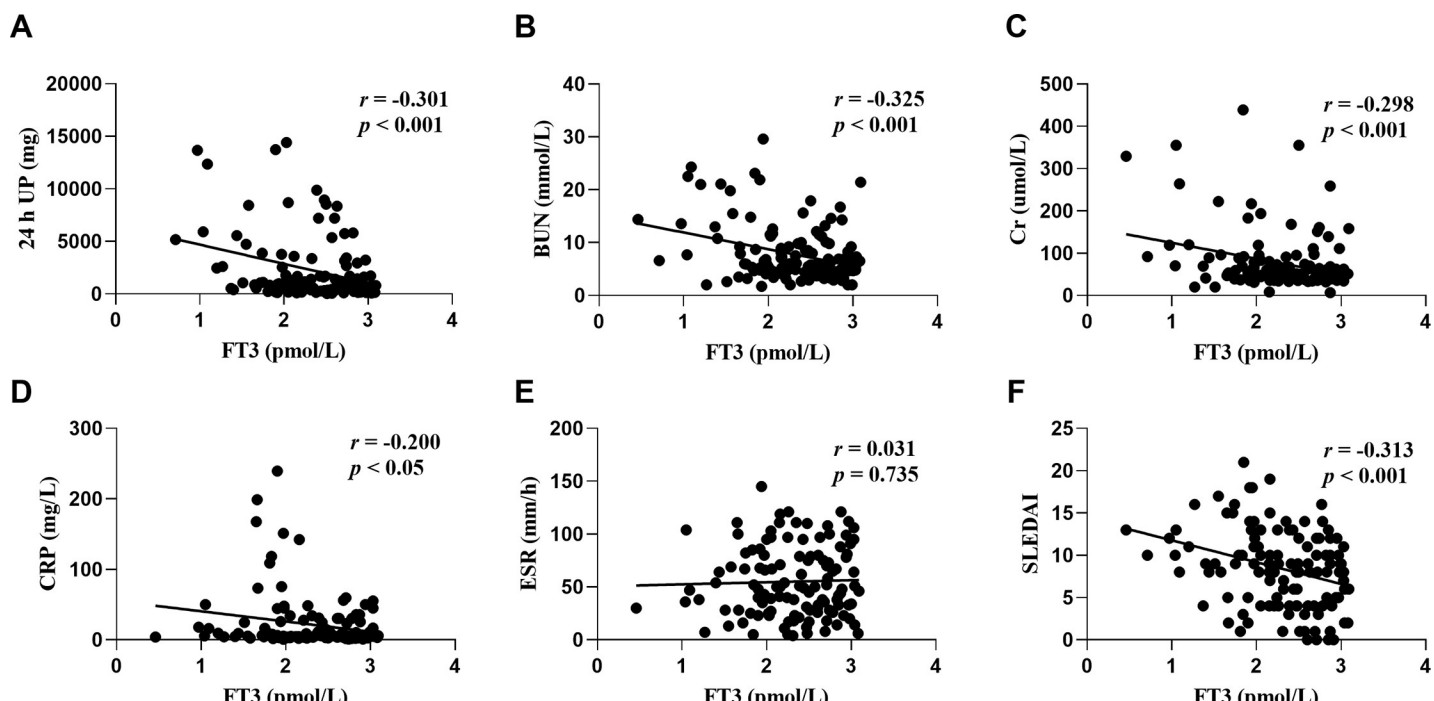

**Fig 1.** Correlation between free triiodothyronine (FT3) and 24 h urine protein (UP) (A), blood urea nitrogen (BUN) (B), creatinine (Cr) (C), C reactive protein (CRP) (D), erythrocyte sedimentation rate (ESR) (E), and systemic lupus erythematosus disease activity index (SLEDAI) (F) in patients with NTIS.

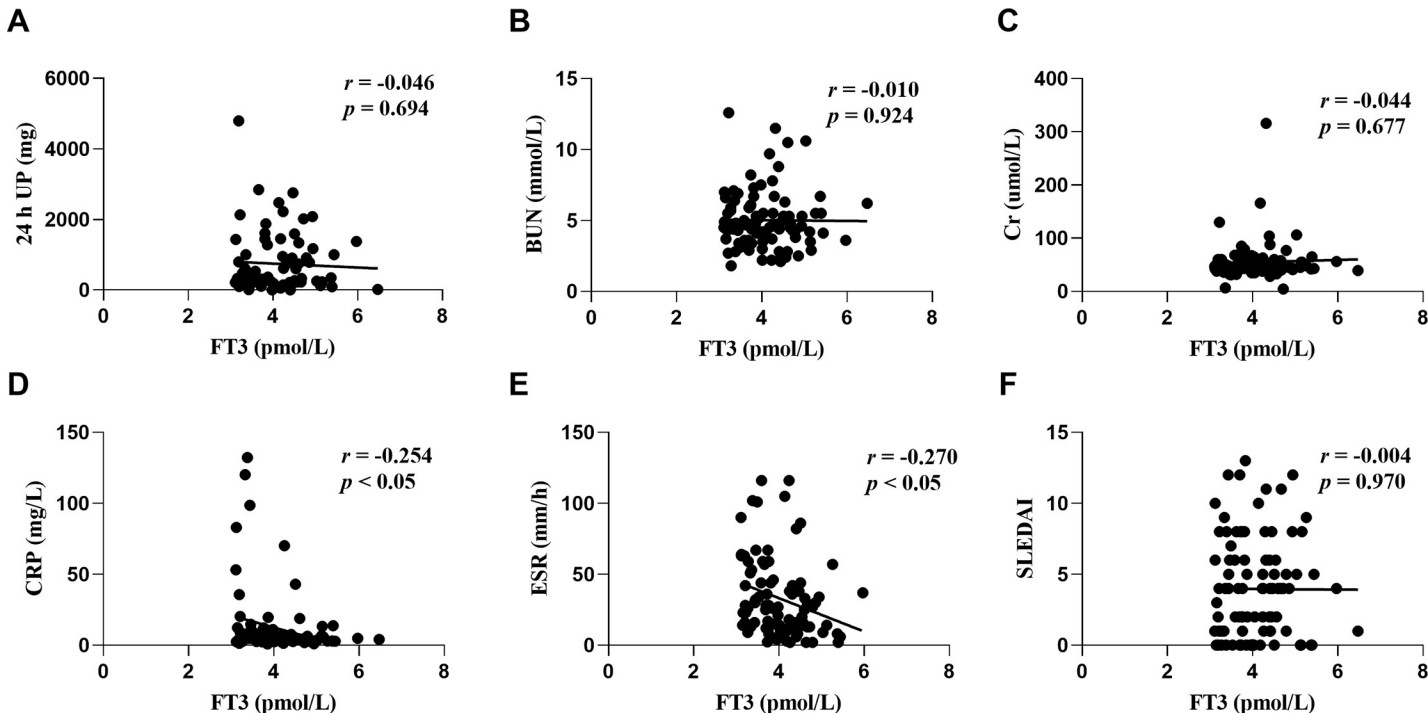

**Fig 2.** Correlation between free triiodothyronine (FT3) and 24 h urine protein (UP) (A), blood urea nitrogen (BUN) (B), creatinine (Cr) (C), C reactive protein (CRP) (D), erythrocyte sedimentation rate (ESR) (E), and systemic lupus erythematosus disease activity index (SLEDAI) (F) in euthyroid patients.

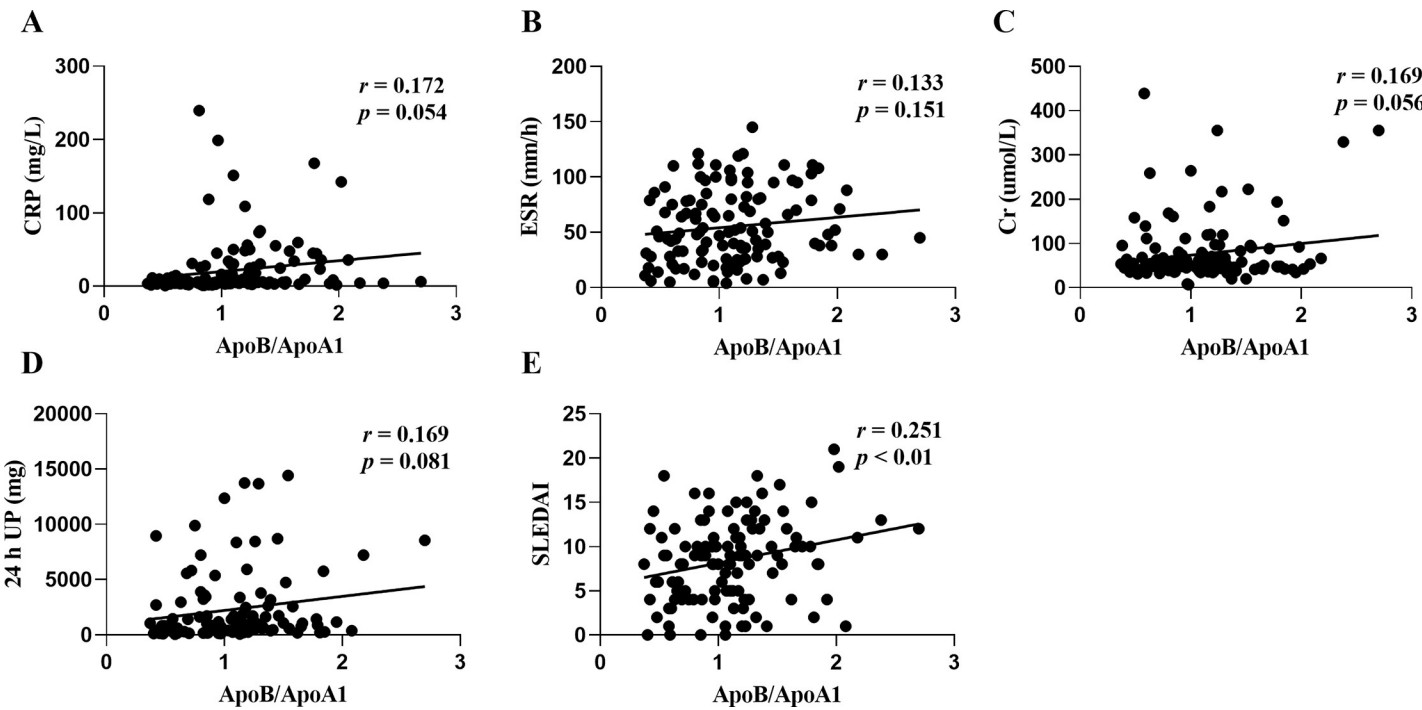

**Fig 3.** Correlation between the atherogenic ApoB/ApoA1 ratio and C reactive protein (CRP) (A), erythrocyte sedimentation rate (ESR) (B), creatinine (Cr) (C), 24 h urine protein (UP) (D), and systemic lupus erythematosus disease activity index (SLEDAI) (E) in patients with NTIS.

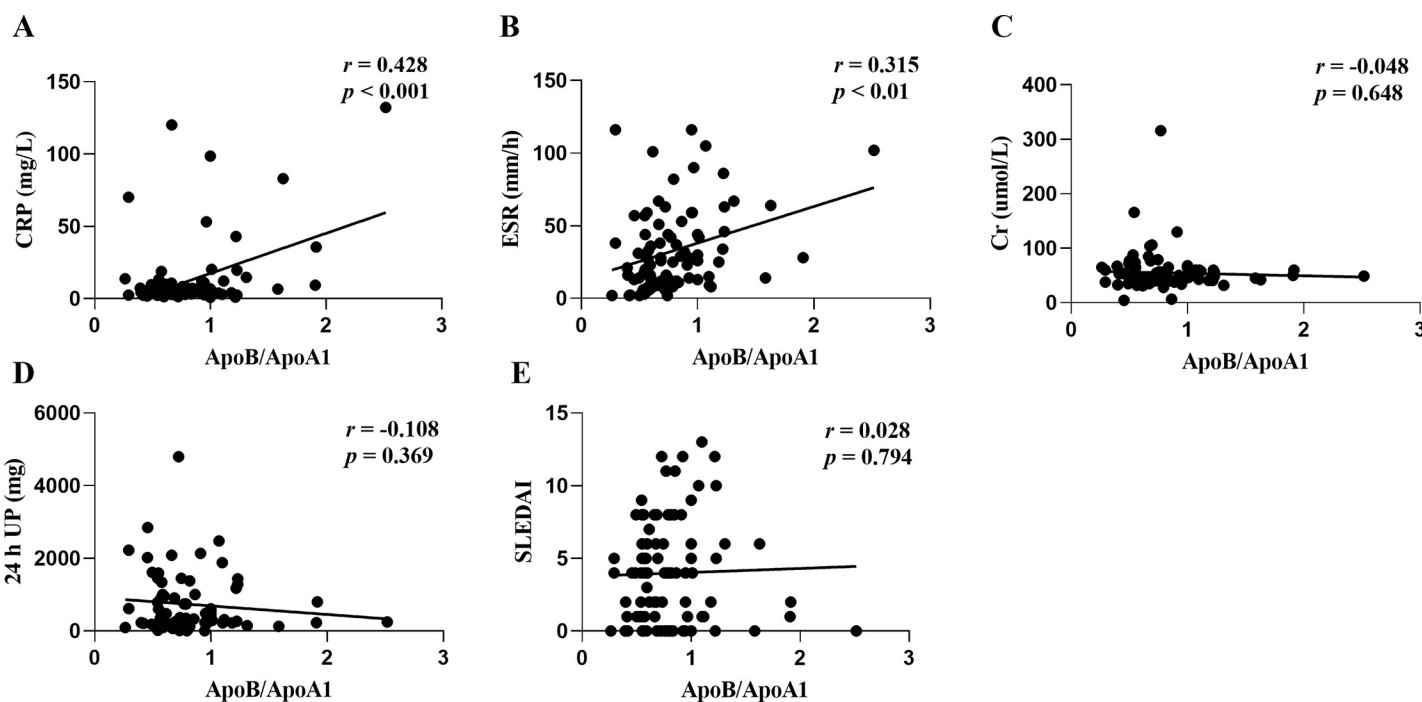

**Fig 4.** Correlation between the atherogenic ApoB/ApoA1 ratio and C reactive protein (CRP) (A), erythrocyte sedimentation rate (ESR) (B), creatinine (Cr) (C), 24 h urine protein (UP) (D), and systemic lupus erythematosus disease activity index (SLEDAI) (E) in euthyroid patients.

significantly associated with CRP ($r = 0.428$; $p < 0.001$; Fig 4A) and ESR ($r = 0.315$; $p < 0.01$; Fig 4B), but not with Cr, 24 h UP and SLEDAI (Fig 4C, 4D and 4F). These results indicate that the atherogenic ApoB/ApoA1 ratio are somewhat related to the SLE disease activity in NTIS patients, as well as immunologic inflammation in euthyroid patients.

Since dyslipidemia was found to be more prevalent in SLE patients with NTIS and correlated with SLE disease-related parameters and poor long-term survival [4], we performed a multivariate logistic analysis to determine whether the level of FT3 was independently associated with the presence of dyslipidemia in patients with SLE. As shown in Table 2, the following factors with a $p$ value less than 0.1 in the univariate analysis underwent a multivariate analysis: FT3, FT4, BUN, CRP, ESR, SLEDAI, and the glucocorticoid dose. Among these factors, only FT3 showed an independent negative association with dyslipidemia ($p = 0.01$; OR = 0.48; 95% CI 0.27 – 0.85).

## Discussion

In the present study, NTIS was found to occur with a high frequency in SLE patients (58.7%). We also observed that the level of FT3 was closely associated with the level of 24 h UP, Cr, BUN, and SLEDAI. These correlations were primarily found in NTIS patients compared to euthyroid patients, which suggests a common pathogenesis between SLE and NTIS. Our results differed somewhat with the study of Colombian patients, in which it was found that autoimmune thyroid disease did not affect SLE severity [14]. An important finding in the present study was that low FT3 was identified as an independent risk factor for dyslipidemia in SLE.

SLE patients have higher blood lipid levels, with an atherogenic lipid profile, termed a 'lupus pattern', which is characterized by elevated concentrations of TC, TG, LDL, and ApoA,

**Table 2. Results of the univariate and multivariate logistic analyses for factors associated with dyslipidemia in patients with SLE.**

| Variables | Univariate analysis | | | Multivariate analysis | | |
|---|---|---|---|---|---|---|
| | *P* | OR | 95% CI | *P* | OR | 95% CI |
| Gender | 0.82 | 0.88 | 0.29–2.63 | | | |
| Age | 0.78 | 1 | 0.98–1.03 | | | |
| FT3 | <0.001 | 0.52 | 0.39–0.70 | 0.01 | 0.48 | 0.27–0.85 |
| FT4 | 0.05 | 0.91 | 0.82–1.00 | 0.71 | 1.03 | 0.88–1.21 |
| TSH | 0.71 | 1.06 | 0.78–1.44 | | | |
| Cr | 0.31 | 1 | 0.99–1.01 | | | |
| BUN | 0.04 | 1.09 | 1.00–1.19 | 0.82 | 1.02 | 0.90–1.15 |
| 24 h UP | 0.33 | 1 | 1.00–1.00 | | | |
| CRP | 0.04 | 1.02 | 1.00–1.04 | 0.97 | 1.00 | 0.98–1.02 |
| ESR | 0.001 | 1.02 | 1.01–1.03 | 0.06 | 1.02 | 0.99–1.03 |
| SLEDAI | 0.08 | 1.07 | 0.99–1.16 | 0.54 | 0.97 | 0.88–1.07 |
| Dose of glucocorticoid | 0.05 | 1.02 | 1.00–1.03 | 0.97 | 1.00 | 0.98–1.02 |

FT3 = free triiodothyronine; FT4 = free thyroxine; TSH = thyroid stimulating hormone; Cr = creatinine; BUN = blood urea nitrogen; 24 h UP = 24 h urine protein;
FGB = fasting blood glucose; CRP = C reactive protein; ESR = erythrocyte sedimentation rate; SLEDAI = systemic lupus erythematosus disease activity index.

as well as decreased HDL [15]. Dyslipidemia is one of the major risk factors for ischemic heart diseases, and several studies have shown a high prevalence of premature atherosclerosis and coronary artery disease in women with SLE [16, 17]. Moreover, lupus nephritis (LN), a common and potentially life-threatening manifestation of SLE, occurs in almost half of all SLE patients [18]. It has been well-established that impaired renal function is associated with dyslipidemia [19, 20]. A study by Kashef *et al.* reported decreased HDL and elevated TG levels in SLE patients with proteinuria of over 0.5 g/day [21]. Moreover, Chong *et al.* determined that there was a negative correlation between TG and the glomerular filtration rate (GFR) in patients with lupus nephritis [22]. In the present study, we found that the ApoB/ApoA1 ratio was significantly correlated with the SLEDAI in NTIS patients and with CRP and ESR in euthyroid patients. There is also increasing evidence to show that treating dyslipidemia is effective for preventing major cardiovascular complications in patients with chronic kidney disease and decreasing the progressive severity of renal failure [23]. Together, these data indicate that dyslipidemia is an important factor associated with renal function, disease activity, and poor prognosis in SLE patients. Thus, the early detection of risk factors and treatment for dyslipidemia is of substantial importance in managing SLE patients; however, the risk factors for dyslipidemia have not been fully elucidated in patients with SLE.

The most important finding of the present study was that FT3, but not SLE disease activity-related parameters (i.e., Cr, BUN, 24 h UP, or SLEDAI), were independently associated with dyslipidemia in SLE. Thyroid hormone plays a critical role in the growth, differentiation, development, and maintenance of metabolic homeostasis [9]. However, the pathogenesis of NTIS remains unknown. Cytokines that are released during illness are known to affect a variety of genes involved in thyroid metabolism and are therefore considered to be a major determinant of NTIS. NTIS is also associated with other illnesses, including severe infections, trauma, major surgery, myocardial infarction, inflammatory conditions, and kidney diseases. Several studies have investigated the effectiveness of thyroid hormone replacement therapy in patients with an underlying disease and NTIS, with controversial results. It was found that synthetic L-T3 replacement therapy significantly improved the neuroendocrine profile and ventricular performance in patients with chronic heart failure [24]. Moreover, short duration

postoperative T3 therapy was shown to increase the cardiac index but did not alter mortality following coronary artery bypass surgery [25]. Although the administration of thyroid hormone in various clinical settings associated with NTIS did not improve the clinical outcome or organ function [26], the duration of replacement therapy was relatively short in most studies. Since the impact of dyslipidemia on the prognosis of SLE continues over a long duration, our results present evidence for the effect of long-term thyroid hormone replacement therapy in patients with SLE and NTIS.

Our study has several limitations. First, this cohort study is from one single center, and a multi-center study would provide more persuasive data. Second, since this is a cross-sectional study, a future long-term prospective study with a larger sample size is required. However, our study is strengthened by the selection of patients who did not have a history of familial hyperlipidemia and/or thyroid disease, and/or diabetes mellitus, and/or other rheumatic diseases, and those who did not take lipid-lowering agents or thyroid medication.

In conclusion, our findings indicate that NTIS frequently occurs in SLE and a low FT3 is associated with SLE disease activity in NTIS patients. Thus, we show for the first time, that a low FT3 is an independent risk factor for dyslipidemia in patients with SLE. Our results suggest that T3 replacement therapy may be useful for improving dyslipidemia and the long-term prognosis of SLE.

## Author Contributions

**Conceptualization:** Zhiyong Chen, Lingyun Sun.

**Data curation:** Xin Zhang, Lirong Liu, Xiaolei Ma, Wei Hu, Xue Xu, Saisai Huang, Bingzhu Hua, Hong Wang, Zhiyong Chen.

**Formal analysis:** Lirong Liu, Saisai Huang, Bingzhu Hua, Hong Wang, Zhiyong Chen.

**Funding acquisition:** Zhiyong Chen.

**Investigation:** Lirong Liu, Xiaolei Ma, Wei Hu, Xue Xu, Saisai Huang, Bingzhu Hua, Zhiyong Chen.

**Supervision:** Hong Wang, Zhiyong Chen, Lingyun Sun.

**Validation:** Zhiyong Chen.

**Writing – original draft:** Xin Zhang, Zhiyong Chen.

**Writing – review & editing:** Zhiyong Chen, Lingyun Sun.

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
