## [Decision Letter · Decision Letter 0]

2 Mar 2020

PONE-D-19-35454

Clinical significance of non-thyroidal illness syndrome on disease activity and dyslipidemia in patients with SLE

PLOS ONE

Dear Dr. Chen,

Thank you for submitting your manuscript to PLOS ONE. After careful consideration, we feel that it has merit but does not fully meet PLOS ONE’s publication criteria as it currently stands. Therefore, we invite you to submit a revised version of the manuscript that addresses the points raised during the review process.

As what recognized by the reviewer, the observation that a correlation of the low FT3 with SLE disease activity in a Chinese population provide some useful insights for a better understudying the disease pathogenesis. However,  important analysis of these correlations in patients with NTIS and in patients without NTIS separately is missing, which is critical to understand whether the association of low FT3 with SLE is associated to NTIS pathogenesis . Please carefully address all reviewer's concerns and the editor will be happy to consider its possible publication. 

We would appreciate receiving your revised manuscript by March 30th. To enhance the reproducibility of your results, we recommend that if applicable you deposit your laboratory protocols in protocols.io, where a protocol can be assigned its own identifier (DOI) such that it can be cited independently in the future. For instructions see: http://journals.plos.org/plosone/s/submission-guidelines#loc-laboratory-protocols

We look forward to receiving your revised manuscript.

Kind regards,

Deyu Fang, Ph.D.

Academic Editor

PLOS ONE

Journal Requirements:

Reviewers' comments:

Reviewer's Responses to Questions

**Comments to the Author**

1. Is the manuscript technically sound, and do the data support the conclusions?

Reviewer #1: Yes

2. Has the statistical analysis been performed appropriately and rigorously? 

Reviewer #1: Yes

3. Have the authors made all data underlying the findings in their manuscript fully available?

Reviewer #1: Yes

4. Is the manuscript presented in an intelligible fashion and written in standard English?

Reviewer #1: Yes

5. Review Comments to the Author

Reviewer #1: The authors performed Pearson's regression model as well as univariate and multivariate logistic regression analyses, and provided correlations between FT3 and UP, GBG, EST, BUN, SLEDAI, Cr, as well as CRP. Thus, they concluded that low FT3 is assosiated with disease activity and an independent risk factor for dyslipidemia in patients with SLE. This article was presented in an intelligible fashion and written in standard English.

6. PLOS authors have the option to publish the peer review history of their article (what does this mean?). If published, this will include your full peer review and any attached files.

Reviewer #1: No

---

## [Author Response · Author response to Decision Letter 0]

13 Mar 2020

Comments from the editor: 

1.This study “Clinical significance of non-thyroidal illness syndrome on disease activity and dyslipidemia in patients with SLE” validated the association between FT3 and SLE disease. They discovered that a low FT3 is associated with SLE disease activity, and a low FT3 is an independent risk factor for dyslipidemia in patients with SLE. This study, while confirmative, provided useful information for FT3 in Chinese population. Despite the authors identified FT3 is a risk factor for dyslipidemia in SLE patients, they didn’t provide the direct correlation analysis between TF3 and biomarkers of lipid profiles.

Reply: Thank you for you critical comments, we revised our manuscript according your advice (see below).

Comments from the reviewers:

1.The authors found that FT3 levels in serum are negatively correlated with the levels of UP, FBG, ESR, BUN, SLEDAI, Cr, as well as CPR. However, the correlations are weak, as shown by the values of correlation coefficients (r) in figure 1. Moreover, it’s unknown how NTIS pathogenesis contributes to these correlations. Therefore, analyzing these correlations in patients with NTIS and in patients without NTIS separately may better address this question. 

Reply: Thank you for you critical comments. We analyzed the associations in patients with NTIS and in patients without NTIS separately. The results showed that the FT3 levels were negatively correlated with 24 h UP (r = -0.301; p < 0.001; Fig. 1A), BUN (r = -0.325; p < 0.001; Fig. 1B), Cr (r = -0.298; p < 0.001; Fig. 1C), as well as SLEDAI (r = -0.313; p < 0.001; Fig. 1F). However, these correlations were not observed in patients without NTIS (Fig. 2A, 2B, 2C and 2F). In addition, FT3 were negatively correlated with CRP in both NTIS patient (r = -0.200; p < 0.05; Fig. 1D) and euthyroid patients (r = -0.254; p < 0.05; Fig. 2D), ESR (r = -0.27; p < 0.05; Fig. 2E) in NTIS patients rather than in euthyroid patients (Fig. 1E). These results indicate that low levels of FT3 are closely related to a higher SLE disease activity mainly in NTIS patients rather than in euthyroid patients. 

SLE is a heterogeneous disease and the aetiology is very complicated. The weak but statistically significant correlations found in this study suggests that NTIS maybe one of the factors impacting the disease activity and long-term prognosis of SLE. 

2.The above concern applies also to relationships between the ApoB/ApoA1 and CRP, ESR, Cr, UP, as well as SLEDAI in figure2. 

Reply: We analyzed the associations in patients with NTIS and in patients without NTIS separately. We found that the ApoB/ApoA1 ratio was significantly associated with SLEDAI (r = 0.251; p < 0.01; Fig.3E) but not with CRP, ESR, Cr, or 24 h UP (Fig. 3A, 3B, 3C and 3D) in NTIS patients. In euthyroid patients, the ApoB/ApoA1 ratio was significantly associated with CRP (r = 0.428; p < 0.001; Fig 4A) and ESR (r = 0.315; p < 0.01; Fig. 4B), but not with Cr, 24 h UP and SLEDAI (Fig. 4C, 4D and 4F). These results indicate that the atherogenic ApoB/ApoA1 ratio are somewhat related to the SLE disease activity in NTIS patients, as well as immunologic inflammation in euthyroid patients.

3.The resolution for Figure1 and Figure2 could be improved.

Reply: We increased the figure resolution to 300 × 300 dpi.

4.The actual number of recruited patients is 223. However, it’s inconsistent in the article. 

Reply：Thank you for your careful review, we corrected this error.

In addition, we deleted the data about fasting blood glucose, because it is neither a criteria of dyslipidemia nor a criteria for SLEDAI.

---

## [Decision Letter · Decision Letter 1]

30 Mar 2020

Clinical significance of non-thyroidal illness syndrome on disease activity and dyslipidemia in patients with SLE

PONE-D-19-35454R1

Dear Dr. Chen,

We are pleased to inform you that your manuscript has been judged scientifically suitable for publication and will be formally accepted for publication once it complies with all outstanding technical requirements.

With kind regards,

Deyu Fang, Ph.D.

Academic Editor

PLOS ONE

Additional Editor Comments (optional):

Reviewers' comments:

Reviewer's Responses to Questions

**Comments to the Author**

1. If the authors have adequately addressed your comments raised in a previous round of review and you feel that this manuscript is now acceptable for publication, you may indicate that here to bypass the “Comments to the Author” section, enter your conflict of interest statement in the “Confidential to Editor” section, and submit your "Accept" recommendation.

Reviewer #1: All comments have been addressed

2. Is the manuscript technically sound, and do the data support the conclusions?

Reviewer #1: Yes

3. Has the statistical analysis been performed appropriately and rigorously? 

Reviewer #1: Yes

4. Have the authors made all data underlying the findings in their manuscript fully available?

Reviewer #1: Yes

5. Is the manuscript presented in an intelligible fashion and written in standard English?

Reviewer #1: Yes

6. Review Comments to the Author

Reviewer #1: The authors fully addressed my concerns, and the added new data support their conclusions. The statistical analysis has been performed appropriately.

7. PLOS authors have the option to publish the peer review history of their article (what does this mean?). If published, this will include your full peer review and any attached files.

Reviewer #1: No

---

## [Editor Report · Acceptance letter]

2 Apr 2020

PONE-D-19-35454R1 

Clinical significance of non-thyroidal illness syndrome on disease activity and dyslipidemia in patients with SLE 

Dear Dr. Chen:

I am pleased to inform you that your manuscript has been deemed suitable for publication in PLOS ONE. Congratulations! Your manuscript is now with our production department. 

With kind regards,

on behalf of

Dr. Deyu Fang 

Academic Editor

PLOS ONE